# Optimal Client Training in Federated Learning with Deep Reinforcement Learning

## Abstract

Federated Learning (FL) is a distributed framework for collaborative model training over large-scale distributed data. Centralized FL leverages a server to aggregate client models which can enable higher performance while maintaining client data privacy. However, it has been shown that in centralized model aggregation, performance can degrade in the presence of non-IID data across different clients. We remark that **training a client locally on more data than necessary does not benefit the overall performance of all clients**. In this paper, we devise a novel framework that leverages Deep Reinforcement Learning (DRL) to optimize an agent that selects the optimal amount of data necessary to train a client model without oversharing information with the server. Starting from complete unawareness of the client's performance, the DRL agent utilizes the change in training loss as a reward signal and learns to optimize the amount of data necessary for improving the client's performance. Specifically, after each aggregation round, the DRL algorithm considers the local performance as the current state and outputs the optimal weights for each class in the training data to be used during the next round of local training. In doing so, the agent learns a policy that creates the optimal partition of the local training dataset during the FL rounds. After FL, the client utilizes the entire local training dataset to further enhance its performance on its own data distribution, mitigating the non-IID effects of aggregation. Through extensive experiments, we demonstrate that training FL clients through our algorithm results in superior performance on multiple benchmark datasets and FL frameworks.

## 1 Introduction

Rise in computational power has enabled learning algorithms to learn from increasingly more data and it has generally been assumed that learning from more data leads to higher performance. However, the amount of data required by the learning algorithm still remains an arbitrary choice driven by personal whim and past experience. At the same time, in distributed systems, continuing to use more data for model training can pose privacy risk concerns, particularly in settings where data can be leaked or used for personal identification (Allouah et al., 2023; Wu et al., 2024; Fowl et al., 2023). Federated Learning has emerged as a powerful framework for distributed learning through which multiple parties, also known as clients, collaborate to train global models without sharing their data Li et al. (2021)McMahan et al. (2017). Centralized FL enables clients to perform limited training on local datasets while the centralized server aggregates the client parameters using different aggregation methods. In this way, each client's data is kept private, and superior performance can be achieved.

Our primary motivation is that training a client locally on more data than necessary does not benefit the overall performance of all clients. This is because the data across different clients are not independent and two sets of data can cancel out their effects on the model update with the aggregation mechanism. Finding the optimal amount of data necessary for local training enables the client to optimize its own performance while maximizing contributions to the global model training through aggregation. Moreover, we empirically find that at the end of the federated learning rounds, the client benefits from unused data in the prior learning rounds by training the final aggregated parameters on the complete local training dataset. This unused data provides the client with fresh information enriching the model parameters. This phenomenon is illustrated through our experiments in Fig. 1.

In this paper, we build a novel Federated Learning framework to find the optimal subset of local training data. We first introduce the notion of an optimal client, which finds the optimal ratio of local training data to train the local model without oversharing local information with the server. To maintain a distinction between the optimal clients and all other clients in the federated learning scheme, we refer to the remaining clients, using all local training data, as naive clients. Selection of the optimal subset is demonstrated in Fig. 1(b) where the radii of the unit circle represent the proportion of data used in naive clients and an optimal client during FL. The annotations on the circumference represent each class in the client's local dataset (CIFAR-10). As shown in Fig. 1(a), using an optimal subset of training data in the optimal client does not hurt the performance of other naive clients. At the same time, our new algorithm can improve the performance of the optimal client compared with the original strategy in FL. To build our optimal client, we introduce Reinforcement Learning during the federated learning rounds to train an RL agent. The RL agent takes the model performance on the client's local dataset in each federated learning round as the current state. An action is defined as changing the optimal ratio of training data to be used for local training. The optimal client treats the federated learning setting as the environment and the reward for the RL

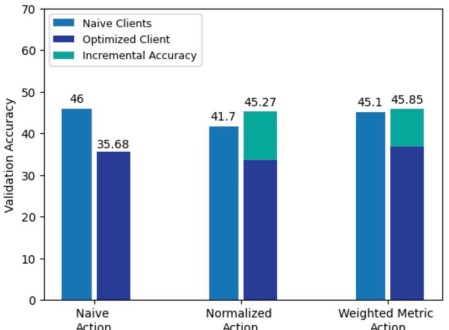

(a) Naive vs. Optimized Accuracy

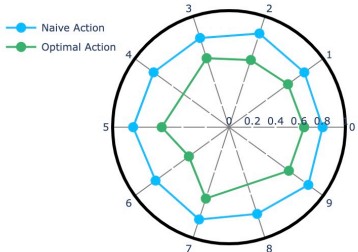

(b) Data selection in each of 10 classes

Figure 1: Naive vs. optimized clients.

agent is the reduction in training loss. The action taken by the agent selects a subset of local data for each class in the dataset. The selected subset is then used by the optimal client for local training to optimize a given metric (i.e., F1 Scores, Recall, Precision, Accuracy, etc.) for each class in the dataset. As the federated learning rounds progress, the agent learns to optimize the amount of local training data used by the optimal client.

The contributions of this paper are summarized as follows:

- We provide a framework based on Deep Reinforcement Learning to select local training data used for a client to be optimized. Additionally, we investigate and present the results of our proposed framework using well-known Federated Learning aggregation algorithms.

- We design two unique functions for the reinforcement learning agent to take actions and adapt them to the existing $\epsilon$-Greedy action selection set up.

- We design a reward function which takes into account the loss of the local client as well as the amount of data utilized in local training.

- We conduct theoretical analysis and proof for an upper bound on the performance of the Optimal Client during Federated Learning.

## 2 PRELIMINARY

**Federated Learning (FL)** is a distributed learning method that preserves data privacy by training models locally on distributed devices. Instead of sharing actual data with a central server, only local models or local model updates are shared. The server implements an aggregation algorithm to combine the local models or model updates into a global model which is then disseminated back to the local clients. A typical FL workflow is presented in Fig. 2. Formally, given a set of $K$ total clients, denote the overall datasets as $D = \{D_1, D_2, ..., D_K\}$ from all clients where each client only leverage its local dataset with $N$ samples $D_k : \{x_n, y_n\}_{n=1}^N$.

In FedAvg McMahan et al. (2017), the federated learning objective can be written as:

$$\min_w f^*(w) \triangleq \frac{1}{K} \sum_{k=1}^{K} f_k(w) \qquad (1)$$

Here $w$ represents the global model parameters and $f_k(w) : \mathbb{R} \mapsto \mathbb{R}$ is the expected local loss of the client defined as $f_k(w) \triangleq \frac{1}{|D_k|} f_k(w, D_k)$ where $f_k$ can be substituted for any loss function. The averaging algorithm can also be replaced by other algorithms such as FedMedian Yin et al. (2018) and FedCDA Wang et al. (2024a).

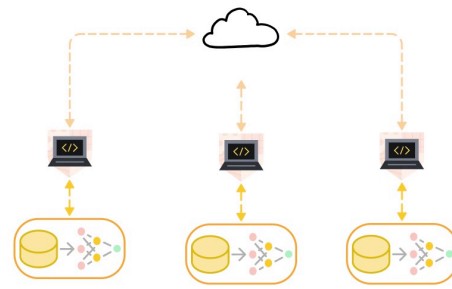

Figure 2: Federated learning workflow.

**Reinforcement Learning (RL)** enables building systems in which agents interact with environments to accomplish one or many tasks. Generally, RL systems are modeled as Markov Decision Processes (MDP). A time step $t$, the agent observes an initial state $S_t \in S$ of the environment. Following a policy $\pi_t(\cdot|s)$, which maps states to actions, the agent takes an action $A_t \in A$. This transitions the environment to the next state $S_{t+1}$ and the agent receives a reward signal $R_{t+1} \in \mathbb{R}$, informing the agent about the quality of its action. As shown in Fig. 3, the agent environment interaction model gives rise to *trajectories* $(S_t, A_t, R_{t+1}, S_{t+1}...)$ (Sutton, 2018). The expected total reward is given as

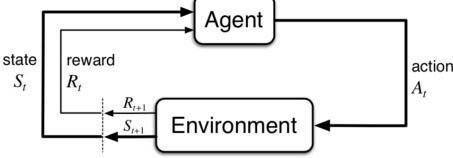

Figure 3: Agent environment interaction.

$G_t = \sum_{k=0}^{\infty} \gamma^k R_{t+k+1}$, where $\gamma \in (0, 1)$ is the discount factor. The value of a given state is meausred by the *State-Value Function* $V_\pi(s) = \mathbb{E}_\pi[G_t|S_t = s]$. Similarly, the quality of an action paired with a state is given by the *Action-Value Function* $q_\pi(s, a) = \mathbb{E}_\pi[G_t|S_t = s, A_t = a]$. The RL objective is to find an optimal policy $\pi_*$ which maximizes an agent's total return. Such a policy shares the optimal state-value function $v_*(s) \doteq \max_\pi v_\pi(s)$ and the optimal action-value function $q_*(s, a) \doteq \max_\pi q_\pi(s, a)$. *Deep Reinforcement Learning (DRL)* combines the function approximation ability of Deep Learning with Reinforcement Learning's sequential decision making. This enables building Reinforcement Learning systems which can generalize to large state and continuous action spaces. Using Deep Learning this process is accomplished by mapping large state spaces to features and features to actions. In recent years, Deep Learning has been extended to Reinforcement Learning methods (Gao et al., 2024; Liu et al., 2024; Schulman et al., 2017; Lillicrap et al., 2016).

## 3 PROBLEM SETUP AND FRAMEWORK

Given the local dataset $D_k$ on a client to be optimized, our target is to generate the optimal amount of training data $D_k^{'}$ for federated learning. Fig. 4 depicts the workflow to optimize the percentage of data in each class for the agent (i.e., the client highlighted with blue) to be optimized. We consider the aggregated parameters of the model on the server as the current state $s_t$. The action $a_t$ is defined as a vector that represents the percentage of samples used for training in each class. Based on the performance of the aggregated parameters on the server, we calculate the reward $r_t$ with a designed reward function. We train the policy $\pi_\theta$ parameterized with $\theta$ based on the reward $r_t$, generated from the loss of the aggregated model $\mathcal{L}_{agg}$ and the loss of the client's local model parameters $\mathcal{L}_l$ based on local training. The training process encourages $\pi_\theta$ to find the optimal percentage of data used in each class for creating $D_k^{'}$ in the upcoming FL rounds. Then, after the process of federated learning, we further leverage the complete dataset $D_k$ to fine-tune the Optimal Client until its performance converges. Using $D_k$ in the final training rounds gives the Optimal Client an incremental boost in performance resulting from unused data in the previous rounds. With the proposed framework, we can not only optimize local training but also guarantee that the data changes on the optimal client have little impact on the performance of other clients.

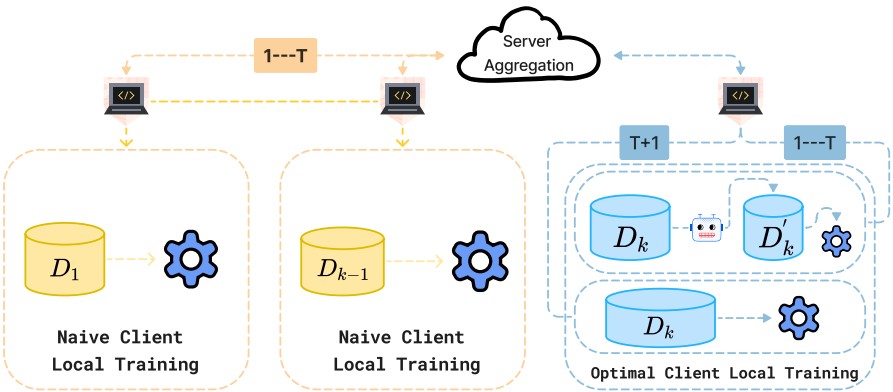

Figure 4: Optimal data selection framework.

## 4 METHOD

Given a total of $T$ federated learning communication rounds, to train the RL agent, we utilize the class-wise performance measured on the local training dataset after server aggregation in the communication round $t$ as the current state $s_t$. In our experiments, we use F1-Score given by F1 $= \frac{2PR}{P+R}$ as a performance measure where $P$ and $R$ are Precision and Recall, respectively Goodfellow et al. (2016)Chinchor & Sundheim (1993). Note that F1-Score can be easily substituted for a different performance measure. The policy consumes this state $s_t$ and outputs a vector action $a_t$ containing weights $z_c$ for each class $c$ in the local dataset.

Formally, for $K$ total clients in the federated learning scheme, with client $k$ as the client to be optimized, $D_k : \{X, Y\}$ as the local dataset for the Optimal Client, and $w_t$ as the server aggregated parameters in the communication round $t$, the state for communication round $t$ is:

$$s_t = \text{F1}(\hat{f}_k(X; w_t), Y) \tag{2}$$

Here, $f_k(w_t)$ is the local model of the Optimal Client parameterized with the server aggregated parameters. Additionally, we implement two action selection strategies and adapt them to the $\epsilon$-Greedy method. Using a parameterized policy $\pi_\theta$, the action in round $t$ is given by:

$$a_t = \pi_\theta(s_t) = [z_1, z_2, \ldots, z_C] \Rightarrow \{z \in \mathbb{R}, b_l \leq z \leq b_u\} \text{ s.t. } b_l, b_u \in (0, 1], \tag{3}$$

where $b_l$ and $b_u$ are user-defined lower and upper bounds respectively.

$\epsilon$-**Greedy Normalized Action** implements a normalized version of the action generated by the policy. The action vector $a_t$ is first normalized and multiplied by the total samples in the local training dataset to get the count of data samples for each class $c$ in the dataset.

$$a'_t \Leftarrow \frac{a_t}{\sum a_t} |D_k| \tag{4}$$

Here $a' = [a'_1, a'_2, \ldots, a'_C]$ is a vector of data sample counts for each class. The class counts are then adjusted to not exceed the total number of samples available for each class. Given that for each class in the local training dataset the maximum class count for each class is $|D_{k_c}|$, and $Unif(0, 1)$ as the *Standard Uniform Distribution* on the interval $(0, 1)$ (Blitzstein & Hwang, 2019), then the $\epsilon$-Greedy Normalized Action is given as:

$$a'_t \Leftarrow \begin{cases} \left[ \frac{max(a'_1, |D_{k_1}|)}{|D_{k_1}|}, \frac{max(a'_2, |D_{k_2}|)}{|D_{k_2}|}, \ldots \frac{max(a'_C, |D_{k_C}|)}{|D_{k_C}|} \right] & \text{if } \frac{1}{\sqrt{t}} < Unif(0, 1), \\ \\ \arg\max_a Q(a) & \text{otherwise.} \end{cases} \tag{5}$$

$\epsilon$-**Greedy Weighted Metric Action** implements a look-back period $\eta$, where every $\eta$ communication rounds, the Optimal Client computes the difference in the absolute value between the current F1-Score and the F1-Score from $\eta$ rounds in the past. The difference is then normalized and for every

class where the F1-Score has decreased since $\eta$ rounds, the weight for that class is increased by the normalized difference. Formally, given $\text{F1}_{t+\eta}$ and $\text{F1}_\eta$ as the F1-Scores in the current round and the F1-Scores from $\eta$ rounds ago, then the normalized difference is given by:

$$\Delta\text{F1} = \frac{|\text{F1}_{t+\eta} - \text{F1}_\eta|}{\sum_c |\text{F1}_{t+\eta} - \text{F1}_\eta|}. \tag{6}$$

With the F1 score, the agent's action is given by:

$$a_t \Leftarrow \begin{cases} a_t + \Delta\text{F1}a_t & \text{if } \frac{1}{\sqrt{t}} < \mathit{Unif}(0, 1), \\[2em] \arg\max_a Q(a) & \text{otherwise.} \end{cases} \tag{7}$$

The Optimal Client utilizes the action generated by the RL policy to create an *Action Partitioned Dataset*, denoted by $D'_k$ such that $D'_k \subset D_k$. Fig. 5 illustrates this procedure.

As the federated learning rounds progress we implement a loss estimation mechanism. Specifically, the Optimal Client waits for $\tau$ communication rounds and then in every subsequent round estimates the local loss for the local training update after the server aggregation in the following round. Based on the assumption that, as the federated learning rounds progress, the client's local training on the local dataset is supposed to produce a lower training loss as we utilize the following piece-wise reward function.

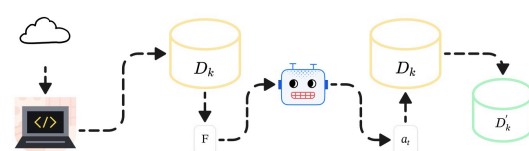

Figure 5: Action partitioned dataset.

$$R_t \Leftarrow \begin{cases} \frac{\mathcal{L}_{\text{agg}} - \mathcal{L}_l}{\mathcal{L}_l} \frac{1}{a_t - \lambda} & \text{if } T < \tau, \\[2em] \frac{\mathcal{L}_{\text{agg}} - \mathcal{L}_{\text{est}}}{\mathcal{L}_{\text{est}}} \frac{1}{a_t - \lambda} & \text{otherwise.} \end{cases} \tag{8}$$

Here, $\mathcal{L}_{\text{agg}}$ is the loss on the local dataset after server aggregation, $\mathcal{L}_l$ on the local dataset after local training, $\mathcal{L}_{\text{est}} = -ue^{-vt}$ is the estimated loss, $a_t = \frac{1}{|a_t|}\sum a_t$ is the mean action generated by the policy $\pi_\theta$, and $\lambda$ is a user defined parameter which normalizes the reward by controlling the amount of local training data generated by the policy. As part of loss estimation, $\mathcal{L}_{\text{est}}$, we fit a non-linear curve (Vugrin et al., 2007) to estimate the parameters, $u$ and $v$, after each federated learning round past $\tau$ rounds. Using equation 2, equation 5, equation 7, and equation 8 we can generate RL trajectories $(s_t, a_t, R_{t+1}, s_{t+1}...)$. The actor-critic paradigm, in Deep Reinforcement Learning, then enables learning a parameterized actor policy $\pi_\theta(a|s)$ which outputs an action $a$ given the current state $s$, as well as a parameterized critic network $v_\varphi(s)$ which approximates a state value function. The critic network can be updated through Mean Squared Error $\nabla\mathcal{L}(\varphi|s_t, a_t) = (\hat{Q}_n(s_t, a_t) - v_\varphi(s_t))^2$ followed by the update for policy network $\nabla_\theta\mathcal{L}(\theta|s_t, a_t) = \hat{Q}_n(s_t, a_t).\nabla_\theta log\pi_\theta(a_t|s_t)$ where $\hat{Q}_n(s_t, a_t) = \sum_{k=0}^{n-1} r_{t+k} + v_\varphi(s_{t+n})$ is the n-step target Plaat (2022).

**Algorithm.** We present the algorithm to train the Optimal Client both from a server as well as a client perspective. The server-side implementation follows a typical federated learning setup up whereas the client-side implementation includes optimized training for the client both during and after Federated Learning. We use DDPG (Deep Deterministic Policy Gradient) (Lillicrap et al., 2016) to train the RL policy. For brevity, we don't include the training of RL policy as part of this algorithm, but details regarding training the RL policy, including the algorithm and the hyperparameters for each experiment, can be found in Appendix A.4.

**Analysis.** In this section, we investigate if the performance of the Optimal Client has an upper bound during the federated learning rounds. Based on the assumption that using more data leads to higher performance, we note that the performance of the Optimal Client will not be as good as

---

**Algorithm 1** Optimal Client Training: $K$(number of total clients), $C \in (0,1) \mapsto \mathbb{R}$ (predetermined ratio of clients to participate in each round), $FederatedAggregation$ (federated learning aggregation algorithm.), $E$ (local train epochs)

---

**Server:**
    initialize $w_0$
    **for** round $t = 0, 1, 2, \cdots, T$ **do**
        $S_t = \{\text{random sample of } C * K \text{ Clients}\}$
        **for** $k \in S$ **in parallel do**
            $w_{t+1} = OptimalClientTrain(k, w_t)$
        **end for**
        $w_{t+1} = FederatedAggregation(S_t)$
    **end for**
**Client:**                             $\triangleright OptimalClientTrain(k, w_t)$
    **while** $t \leq T$ **do**
        Compute $s_t$ using Equation. 2
        Compute $a_t$ using Equation. 3
        $D_k^{'} \leftarrow D_k(a_t)$                 $\triangleright ActionPartitionedDataset$
        $B \leftarrow \{\text{Create batches of size } B \in D_k^{'}\}$
        **for** $e = 1, 2, 3 \cdots$ **in** $E$ **do**
            **for** $b$ **in** $B$ **do**
                $w_t \leftarrow w_t - \eta \nabla l(w; b)$
            **end for**
        **end for**
    **end while**
    return $w_t$ to server
    $B \leftarrow \{\text{Create batches of size } B \in D_k\}$
    **for** $e = 1, 2, 3 \cdots$ **in** $E$ **do**                     $\triangleright UntilConvergence$
        **for** $b$ **in** $B$ **do**
            $w_t \leftarrow w_t - \eta \nabla l(w; b)$
        **end for**
    **end for**

---

if it was trained on its entire local dataset. Under this assumption we elucidate the answer to one main question: *Is there an upper bound to the performance of the Optimal Client during Federated Learning*.

**Proposition** (Performance Bound of Client Training) Given $s_t$ and $a_t = [z_1, z_2, \ldots, z_C] \Rightarrow \{z \in \mathbb{R}, 0 \leq z \leq 1\}$ as the state and the action taken by the policy, let $z$ be the radius of a unit circle representing the total available sample size for each class in the *Action Partitioned Dataset* $D_{k_{1,2,3\cdots C}}^{'} \forall c \in C$. Let $A = [Z_1, Z_2, \ldots, Z_C] \Rightarrow \{Z \in \mathbb{R}, 0 \leq Z \leq 1\}$ be a vector representing the total samples for each class in the complete local client dataset $D_{k_{1,2,3\cdots C}}^{'} \forall c \in C$. Additionally, let $\omega = Z_C^2 - z_c^2$ be the difference in the squared radii. The performance bound, of the client trained on the complete dataset, for class $c$ is defined as the area of the circle:

$$P_{k_c} = \pi Z_c^2 \tag{9}$$

Using Equation 9, the total performance of client $k$, on the complete local dataset, is given as:

$$P_k = \pi Z_1^2 + \pi Z_2^2 + \pi Z_3^2 + \cdots + \pi Z_c^2$$

Similarly, using Equation 9, the performance of client $k$ on the *Action Partitioned Dataset* is:

$$P_k^{'} = \pi z_1^2 + \pi z_2^2 + \pi z_3^2 + \cdots + \pi z_c^2$$

**Theorem:** *A client trained on the Action Partitioned Dataset $D_k^{'}$ relative to the entire local dataset $D_k$ has a performance bound given by:*

$$P_k - P_k^{'} \leq \Omega$$

*Proof:*

$$P_k - P_k^{'} = \pi Z_1^2 + \pi Z_2^2 + \pi Z_3^2 + \cdots + \pi Z_c^2 - \pi z_1^2 - \pi z_2^2 - \pi z_3^2 - \cdots - \pi z_c^2$$
$$= \pi Z_1^2 - \pi z_1^2 + \pi Z_2^2 - \pi z_2^2 + \pi Z_3^2 - \pi z_3^2 + \cdots + \pi Z_C^2 - \pi z_C^2$$
$$= \pi(Z_1^2 - z_1^2) + \pi(Z_2^2 - z_2^2) + \pi(Z_3^2 - z_3^2) \cdots \pi(Z_C^2 - z_C^2)$$
$$= \pi \omega_1 + \pi \omega_2 + \pi \omega_3 \cdots \pi \omega_C$$
$$\leq \pi \sum_{c=1}^{C} \omega_c = \Omega$$

The proof above shows that constructing a dataset $D_k^{'}$ by minimizing the difference between the action taken by the policy and the total sample size for each class in the dataset will lead to better performance by the Optimal Client during the federated learning rounds. However, we note that this also represents a trade-off between the performance improvement that the Optimal Client will benefit from by retaining these data samples for training post the federated learning rounds.

## 5 EXPERIMENTAL SETUP

**Methods.** We conduct experiments of our proposed methodology using 5 Federated Learning aggregation baseline algorithms. These algorithms include FedCDA Wang et al. (2024a), FedProx Li et al. (2020), FedMedian Yin et al. (2018), FedAvgM Hsu et al. (2019), and FedAvg McMahan et al. (2017). For Deep Reinforcement Learning we use DDPG Lillicrap et al. (2016) and we use ResNet50 He et al. (2016) as the server and the client models.

**Datasets.** We conduct our experiments using 3 datasets including, CIFAR 10, CIFAR 100 Krizhevsky et al. (2009), and FashionMNIST Xiao et al. (2017). Each dataset contains 10, 100, and 10 classes, respectively. From the overall dataset, we create non-IID partitions using the Dirichlet Partitioner Yurochkin et al. (2019), and each partition is given to each client as its own local dataset. Furthermore, each partition is split using 80/20 training and validation split, where 80% of the data is used for training and 20% of the data is used for validation.

**Results and Analysis** Our experiments show the utility of our proposed method compared to well-established baselines. We conduct 100 Federated Learning rounds for 8 clients where each client is trained for 1 local epoch. Through our experiments, we demonstrate the generalization capability of our method in different federated learning settings. The results from our experiments are summarized in Table 1, where we show a comparison of the best mean performance of the Naive Clients, including Precision, Recall, and Accuracy, relative to the Optimal Client on the validation datasets. Each two-row combination represents a comparison of the mean performance achieved by all naive clients in the federated learning setup relative to the best performance of the Optimal Client achieved after training on the complete local training dataset, $D_k$, post the federated learning rounds.

Fig. 6 shows the mean validation accuracy of the Naive Clients relative to the Optimal Client, after each server aggregation during the federated learning rounds. The final validation accuracy of the Optimal Client is plotted as a separate line which shows the best validation accuracy attained by the Optimal Client by training on its entire local dataset after the federated learning rounds. It can be observed that the Optimal Client produces lower performance relative to the naive client during the federated learning rounds. This phenomenon is illustrated in Fig. 7 and attributed to the fact that during the federated learning rounds the Optimal Client utilizes a smaller proportion of the local dataset relative to all other clients. Fig. 7(a) shows normalized actions and Fig. 7(b) shows weighted metric actions, taken by the RL policy in comparison to naive data selection using 80/20 train test split. During the federated learning phase, the RL policy determines the minimum viable amount of data necessary for local training. However, after the federated learning rounds finish, the Optimal Client is trained on its entire local dataset until it converges. During this phase of local training, the Optimal Client exhibits superior performance. In addition to the performance improvement of the Optimal Client post federated learning rounds, we also observe a considerable increase in convergence speed which can be ascribed to the fact that the Optimal Client resumes local training using the aggregated parameters from the final federated learning round.

**Ablation Study.** As part of our ablation study, we conduct experiments using naive actions for every client. Naive actions correspond to each client's dataset being split using the 80/20 split. The results

| | Cifar 10 | | | FashionMNIST | | | CIFAR 100 | | |
|---|---|---|---|---|---|---|---|---|---|
| | Precision | Recall | Accuracy | Precision | Recall | Accuracy | Precision | Recall | Accuracy |
| FedAvg | 50.28 | 38.15 | 29.65 | **73.41** | 52.58 | 37.47 | 16.88 | 16.15 | 13.01 |
| FedAvg + Our Method | **52.41** | **48.28** | **43.22** | 67.23 | **58.19** | **53.31** | **20.17** | **17.99** | **24.90** |
| FedAvgM | 50.27 | 38.00 | 31.10 | **76.04** | 55.24 | 38.14 | 16.90 | 16.13 | 13.26 |
| FedAvgM + Our Method | **53.96** | **48.85** | **43.36** | 67.24 | **58.69** | **51.25** | **20.87** | **18.61** | **26.10** |
| FedMedian | 44.20 | 35.48 | 31.81 | **75.42** | 56.45 | 42.67 | 15.84 | 15.40 | 13.21 |
| FedMedian + Our Method | **53.72** | **47.28** | **42.58** | 64.45 | **59.29** | **49.74** | **20.82** | **19.02** | **25.75** |
| FedProx | 50.84 | 39.10 | 31.20 | **75.92** | 54.93 | 37.64 | 16.73 | 16.00 | 13.27 |
| FedProx + Our Method | **53.51** | **48.50** | **42.94** | 65.90 | **57.66** | **50.18** | **21.34** | **18.42** | **26.14** |
| FedCDA | 46.52 | 34.42 | 30.69 | **76.38** | 55.54 | 38.56 | 13.44 | 12.95 | 12.10 |
| FedCDA + Our Method | **57.03** | **50.61** | **43.64** | 66.19 | **56.99** | **49.87** | **21.54** | **19.50** | **26.47** |

Table 1: Performance comparison with baseline methods. Each two-row combination shows the mean performance of naive clients, over the complete federated learning rounds, relative to the performance of the Optimal Client, after the federated learning rounds, from training on the complete local dataset.

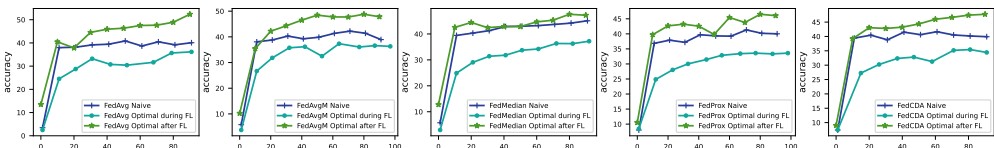

Figure 6: Mean accuracy in FL rounds. The blue line represents the mean accuracy of all naive clients. The green line represents the accuracy of the Optimal Client. The dark green line represents the accuracy of the Optimal Client in each epoch after federated learning rounds.

of our ablation experiments are summarized in Table 2. It is evident from the results that learning a RL policy to partition the local dataset, during the federated learning rounds, followed by training on the complete local dataset, yields improved overall performance for the Optimal Client.

**Discussion.** Our experimental findings show that training a client on a subset of its own local data allows the client to improve its performance during the federated learning rounds, and benefit considerably by training on the complete dataset after the federated learning rounds. Utilizing RL, a parameterized policy can be learned and optimized, on the client's local performance, as the client interacts with the server. This enables the client to dynamically create subsets of its local training data. During federated learning, the client benefits from aggregation while post federated

| | Precision | Recall | Accuracy |
|---|---|---|---|
| FedAvg (original) | 78.96 | 59.10 | 41.21 |
| FedAvg (with optimal client) | 73.41 | 52.58 | 37.47 |
| FedAvgM (original) | 77.45 | 56.49 | 38.86 |
| FedAvgM (with optimal client) | 76.04 | 55.24 | 38.14 |
| FedMedian (original) | 76.68 | 56.92 | 41.28 |
| FedMedian (with optimal client) | 75.42 | 56.54 | 42.67 |
| FedProx (original) | 78.70 | 58.68 | 40.62 |
| FedProx (with optimal client) | 75.92 | 54.93 | 37.64 |
| FedCDA (original) | 73.92 | 52.74 | 37.19 |
| FedCDA (with optimal client) | 76.38 | 55.54 | 38.56 |

Table 2: Effects of the optimal client on other naive clients. All experiment was conducted on the local dataset using 80/20 training and validation split.

learning the client leverages information from unused samples to further improve its performance. We note that training on a smaller subset of data can make the Optimal Client marginally lag in performance relative to other clients. This sets up our motivation to further investigate potential solutions for maintaining competitive performance during the federated learning rounds.

## 6 RELATED WORKS.

Since our work prioritizes improving client performance in a federated learning setting, we provide an overview of related methods and techniques that address data heterogeneity issues and improve client personalization. These areas form the cross-section of technologies that enable our research.

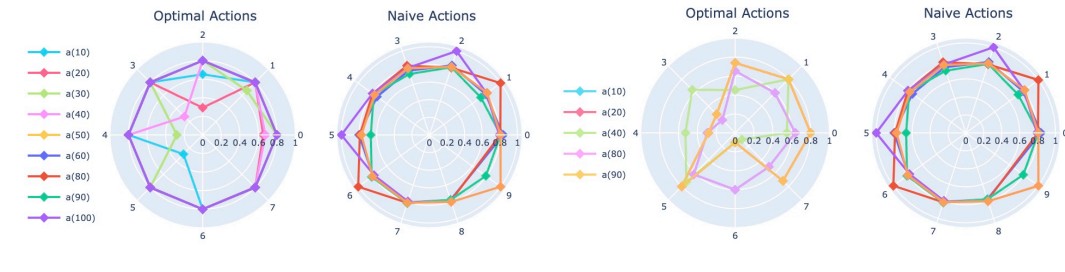

(a) Normalized vs. Naive Actions        (b) Weighted Metric vs. Naive Actions

Figure 7: Optimal actions taken by the RL agent in different federated learning rounds, versus Naive Actions taken by each client. The legend displays actions taken by the RL agent.

**Data Heterogeneity Issues.** Data Heterogeneity can potentially have an adverse impact on model convergence as well as final model performance (Kim et al., 2023; Yu et al., 2023; Heinbaugh et al., 2023; Li et al., 2020; Karimireddy et al., 2020). To address this issue, many variants of FL aggregation algorithms, since FedAVG McMahan et al. (2017) have been proposed. FedProx Li et al. (2020) add a proximal term to get the local models to be closer to the global model. FedDC Gao et al. (2022) addresses data heterogeneity through a local drift variable which improves model consistency and performance, resulting in faster convergence across diverse tasks. FedCDA Wang et al. (2024a) addresses this issue in a cross-round setting by selecting and aggregating local models that minimize divergence from the global model. Tang et al. (2024) improve client updates in an attempt to improve the global model performance. Huang et al. (2024) introduce two compressed FL algorithms that attain improved performance under arbitrary data heterogeneity. (Wang et al., 2024b; Li et al., 2024) study data heterogeneity in an asynchronous setting and propose methods for caching local client updates to measure each client's contribution to the global model as well as reducing staleness of clients in global model updates.

**Personalization and Optimization.** Due to device as well as data heterogeneity, training client models on local data can potentially result in better outcomes relative to participating in federated learning (Wu et al., 2020). Personalization (Xu et al., 2023; Tan et al., 2022) attempts to circumvent this shortcoming by improving client performance while taking local data distribution of a client into consideration (Jiang et al., 2024). Huang et al. (2021) propose a method, FedAMP, by which they enable a message passing mechanism between similar clients in a federated setting to improve performance amongst them. FedALA Zhang et al. (2023) achieves better personalization by adapting to the local objective through element-wise aggregation of the global and the local model. FedPAC Scott et al. (2024) implements a regularization term to account for the label distribution shift scenario amongst clients, and learns shared feature extraction layers in deep neural networks across clients as well as shared classification heads in clients with similar data distributions. (Wang et al., 2024c; Kim et al., 2024; Cheng et al., 2024) study hyperparameter optimization and momentum to gain faster convergence whereas (Fan et al., 2024) study client fairness based on contribution. Chanda et al. (2024) strive for improved performance by training clients on coresets of their local training data, by assigning a weight vector to each client, which acts as the coreset weight.

## 7 CONCLUSION

In our work, we propose a novel method to train clients for improved personalization through efficient usage of the client's own local data. In doing so, we leverage deep reinforcement learning's planning and sequential decision making capabilities. Our method shows that efficient utilization of local data can enable clients to have better performance compared to naive training on the local dataset during federated learning. Additionally, we show that a learned RL policy, by designing an adequate reward function, can aid the client in optimizing its performance. We note that utilizing a smaller subset of local data can result in lower performance during the federated learning rounds and to remedy this we establish a theoretical upper bound on client performance, and present a trade-off between improving performance during federated learning rounds versus improving performance post federated learning. Overall, we hope that our work encourages more research interest in utilizing RL to orchestrate client training in a federated setting and future works extend the ideas presented in our work to multiple clients using multi-agent as well as model-based RL systems.

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

# A APPENDIX

## A.1 PRECISION, RECALL, AND F1-SCORES

Based on the formulations in (Japkowicz & Shah, 2011), given a classifier $f$, Precision (P), Recall (R), and F$\alpha$-Scores (F$\alpha$) are defined as:

$$P(f) = \frac{\text{TP}}{\text{TP} + \text{FP}}$$

$$R(f) = \frac{\text{TP}}{\text{TP} + \text{FN}}$$

$$F\alpha(f) = \frac{(1 - \alpha)(P(f) * R(f))}{\alpha P + R}$$

As a variant of F-Scores, with $\alpha = 1$, F1-Score (F1) is defined as:

$$F1(f) = \frac{2(P(f) * R(f))}{P + R}$$

Where TP, FP, and FN, are True Positives, False Positives, and False Negatives, respectively.

## A.2 VALIDATION PLOTS

The validation accuracy plots for each dataset including Fashion Mnist, CIFAR 10, and CIFAR 100 are presented below.

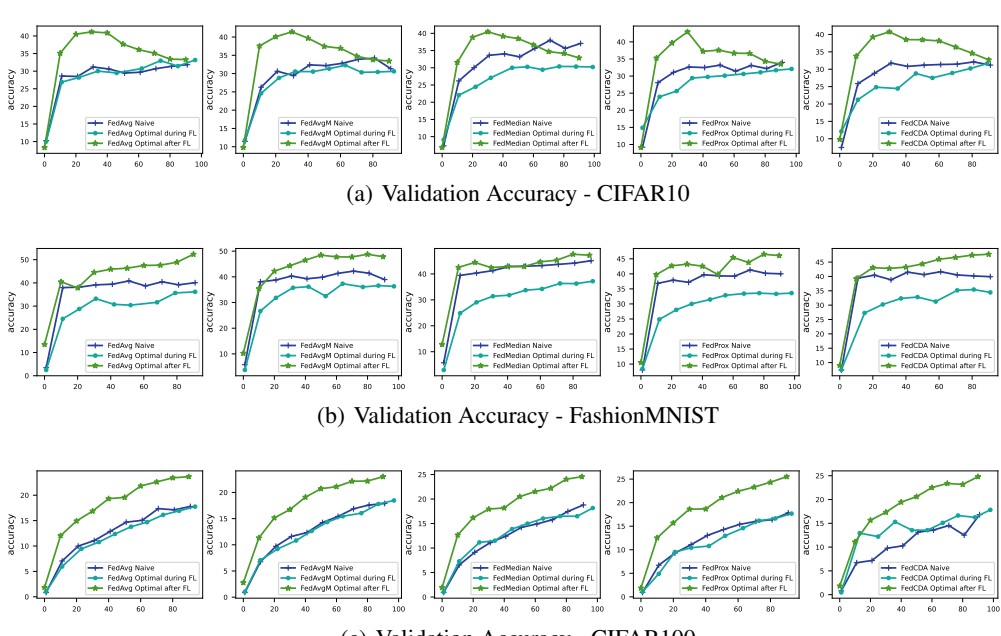

(a) Validation Accuracy - CIFAR10

(b) Validation Accuracy - FashionMNIST

(c) Validation Accuracy - CIFAR100

## A.3 EXPERIMENT HYPERPARAMETERS

The hyperparameters for the federated learning procedure are given below:

```
NUM_CLIENTS = 8
LOCAL_TRAINING_EPOCHS = 1
LOCAL_LEARNING_RATE = 1e-5
LOSS_ESTIMATION_WAITING_PERIOD = 5
LOCAL_TRAINING_BATCH_SIZE = 16

DATASETS = [
    {
        'name': 'cifar100',
        'num_classes': 100,
        'input_shape': 224,
        'training_periods': 100,
        'optimizer_config':
            {
                'learning_rate': 0.001,
                'learning_rate_decay': 0.1,
                'learning_rate_decay_period': 30,
                'weight_decay': 1e-4,
            },
    },
    {
        'name': 'cifar10',
        'num_classes': 10,
        'input_shape': 224,
        'training_periods': 100,
        'optimizer_config':
            {
```

```
                        'learning_rate': 0.001,
                        'learning_rate_decay': 0.1,
                        'learning_rate_decay_period': 30,
                        'weight_decay': 1e-4,
                },
        },
        {
            'name': 'fashion_mnist',
            'num_classes': 10,
            'input_shape': 224,
            'training_periods': 100,
            'optimizer_config':
                {
                    'learning_rate': 0.0001,
                    'learning_rate_decay': 0.1,
                    'learning_rate_decay_period': 30,
                    'weight_decay': 1e-4,
                },
        }
]

#retraining the Optimal Client after the federated learning rounds
RETRAINING_LEARNING_RATE = 1e-6
```

## A.4 RL TRAINING

RL training is conducted, in an episodic manner, using DDPG (Deep Deterministic Policy Gradient) (Lillicrap et al., 2016) adapted to continuous actions using (Lapan, 2020). In the actor and the critic networks we use $Softplus$ activation. Both networks are optimized using Stochastic Gradient Descent (SGD) (Ruder, 2016) with a Cosine Annealing Learning Rate scheduler (Loshchilov & Hutter, 2016). Hyperparameters for the training procedure are presented below:

```
GAMMA = 0.99 #reward discount factor
REWARD_STEPS = 4
EPISODE_LENGTH = 4

#number of hidden neurons in the actor and critic networks
HID_SIZE = 128

#SGD learning rate
ACTOR_LEARNING_RATE = 0.02
CRITIC_LEARNING_RATE = 0.05
```

## A.5 ENVIRONMENT AND LIBRARIES.

Our experiments are implemented in Python. Additionaly, we use scientific programming libraries including scikit-learn Buitinck et al. (2013), Numpy Harris et al. (2020), Flower Beutel et al. (2020), Scipy Virtanen et al. (2020), and PyTorch Paszke et al. (2019). All plots are generated using Matplotlib Hunter (2007) and Plotly Inc. (2015). The experiments are conducted using 3 NVIDIA GeForce RTX 3080 GPUs.

