# OpenReview forum: "Optimal Client Training in Federated Learning with Deep Reinforcement Learning"
_ICLR.cc/2025/Conference — ICLR 2025 Conference Withdrawn Submission_

### Official Review · Reviewer_oBVS · 2024-11-01

**Soundness:** 1
**Presentation:** 1
**Contribution:** 2
**Rating:** 1
**Confidence:** 4

**Summary:**

This paper introduces a DRL-based framework to optimize data selection for client model training, reducing unnecessary data sharing with the server. The DRL agent uses changes in training loss as a reward to adjust data volumes, thereby improving client outcomes. After each aggregation round, the DRL algorithm considers the local performance as the current state, and the agent selects optimal class weights for the next training cycle, learning a policy for effective dataset partitioning in federated learning. Finally, each client leverages its entire dataset to fine-tune on its data distribution, enhancing personalization.

**Strengths:**

Using reinforcement learning to optimize the amount of data per client in federated learning.

Provide some empirical and theoretical analysis of the proposed method.

**Weaknesses:**

In my opinion, the presentation of this work does not meet ICLR standards. This can be observed in several ways, which I will discuss briefly below:

Misleading title: The term "optimal" is mentioned in the title without any proof of optimality. Such a strong claim should only be made when supported by provable theoretical optimality or extensive experimental analysis demonstrating this.

Figure 3 is directly taken from Richard Sutton’s Introduction to RL book, without any modifications or credit given to the original source. This constitutes plagiarism. The authors must update the figure and provide proper attribution.

The main text could easily be condensed to 6 pages. The first five figures take up too much space; some should be moved to the appendix (such as Figures 2 and 3), some should be combined (such as Figures 4 and 5), and others should be reduced in size (such as Figures 1 and 4). General figures could be placed in an appendix, while priority should be given to explaining and discussing the proposed method.

Additionally, Algorithm 1 occupies excessive space unnecessarily, which should be avoided. Several paragraphs should also be condensed, including the contributions section, which in their case does not require four separate bullet points. Furthermore, the preliminary section (Section 2) is unnecessary and could be condensed within the introduction. If the authors wish, they could place this section in the appendix.

The second paragraph of the introduction, in addition to its length, discusses the results in Figure 1, which are presented too early and include undefined concepts such as "normalized action" and "weighted metric action."

There are repetitive definitions of the term RL throughout the paper, where both "reinforcement learning" and "Reinforcement Learning" are used inconsistently. The same issue arises with "FL," which is defined but inconsistently replaced by "Federated Learning" and "federated learning." The authors should define each term once, for example by writing "Reinforcement Learning (RL)" and then consistently using "RL" throughout the entire paper.

In the experimental section, the results are based on only one repetition, without any additional runs. This makes the results unreliable. The authors should conduct multiple repetitions and report the means and standard deviations to improve the robustness of the findings.

In the experimental section, the results are conducted for 8 clients, a diverse and larger number of clients should be considered in this study.

**Questions:**

N/A

---

### Official Review · Reviewer_ok9y · 2024-11-01

**Soundness:** 2
**Presentation:** 2
**Contribution:** 2
**Rating:** 3
**Confidence:** 5

**Summary:**

This paper presents an approach within Federated Learning (FL) to optimize the subset of local data used by clients during model training based on unbalanced data. They introduce a DRL-based framework, where an RL agent dynamically adjusts the data usage ratio for optimal clients. The framework distinguishes between optimal clients, who use minimal effective data, and "naive" clients who use all available data, demonstrating that optimized data usage does not detract from overall model performance.

**Strengths:**

Controlling the sampling rate is a known approach to cope with unbalanced data. The idea of using RL to find the ratios is a nice idea.

**Weaknesses:**

The idea is worth additional work. The authors work is very superficial: the state in RL is single dimensional which clearly begs for more sophisticated state representations.

The paper is not written well. There are many issues.
- Section 3 first states that the state corresponds to the aggregated parameters, but then it is actually only the performance metric.
- Using the term "Optimal Client" is very confusing and it is unclear what is the association with optimality.
- When specifying actions, z_1, ... z_C are actually not used when imposing the lower and upper bounds

The handling of the policy is unclear. Apparently it is a parametric policy but its form is never specified.
In DDPG the Q-factor is used but advantage is more frequent.

The two theorems are very unclear. In the proposition it is not clear at all what is actually being stated. It seems to include only assumptions.
The theorem is using Omega which seems to be crucial but it is never defined.
I can find its definition in the proof but that is a little bit too late.

The ablation study is actually not an ablation study. It is a robustness study or sensitivity with respect to data split.

There are many typos: \hat{f}_k and f_k are actually the same but the notation is different; in (8) T should be t, etc

**Questions:**

1 Why using Q and not advantage?
2 why aren't all clients using the RL strategy? In other words, each client should be using it?
3 What kind of a parametric policy is being used?
4 Why is each client trained only for 1 epoch?

---

### Official Review · Reviewer_DARN · 2024-11-02

**Soundness:** 1
**Presentation:** 1
**Contribution:** 2
**Rating:** 3
**Confidence:** 4

**Summary:**

This work develops a deep reinforcement learning (DRL) algorithm to adjust local training data in federated learning (FL). It treats the metric scores (e.g., F1 scores) of each class of the current model as the state Eq.(2), and uses a policy to select the portion of data to be used for each class Eq.(3).

**Strengths:**

This work provides an intuition that using all data from a client may not be the best option and develops a DRL algorithm to select the portion of data to use, which can achieve some empirical success.

**Weaknesses:**

1. The writing of the paper can be greatly improved in terms of readability and preciseness. For example,

- Fig.1(a) is hard to interpret without knowing the details later.
- The term “optimal client” is misleading. It seems to mean optimized client instead.
- L161 mentions that the algorithm can “guarantee that the data changes on the optimal client have little impact on the performance of other clients”, which seems to be an empirical claim at best without any theoretical justification.

2. Several algorithm designs are questionable.

2.1. The $\epsilon$-greedy normalized cation in L198 is very questionable

- If one wants the normalization in Eq.(4), it can be achieved by using a simple softmax for the policy network instead of using Eq.(3). Also, the summation in (4) should be the L1 norm for clarity.
- More importantly, Eq.(5) is not $\epsilon$-greedy, at least in the RL context. $\epsilon$-greedy is taking the argmax most of the time and takes a *uniformly random action* in the $\epsilon$ case. Here in Eq.(5), the $\epsilon$ case (first case) actually depends on the policy network (as it depends on the calculated actions). Also, shouldn’t the max be min instead?
- Finally, how is the argmax computed in the second case of Eq.(5)? The action here is continuous, so maximizing Q-value is a hard optimization problem that needs to be solved in each RL step.

2.2. For the weighted metric action has similar problems in Eq.(7), and it has additional problems:

- L217 says it is “formally” given by (6), while this equation is not very formal (or mathematically accurate). Assuming that the F1 scores in Eq.(6) is a vector. Its denominator should be the L1 norm of the difference. Additionally, it does not reflect L216, choosing only the classes with decreased performance.
- In the first case of (7), the action may be out of bounds after adding the delta F1 scores, which is not handled explicitly in the paper.

2.3. Eq.(8) is not well-motivated and the RL training requires more discussion.

- Why do we need to switch after $\tau$?
- Why should the estimated loss use the form specified in L250?
- Why is the same $a_t$ used on both sides of the equation in L250 but they mean different things?
- Why there is a $\nabla$ in L258?
- What are the REWARD_STEPS and EPISODE_LENGTH in L788 and 789?

3. The theoretical analysis is not very meaningful.

- L269 "Based on the assumption that using more data leads to higher performance, we note that the performance of the Optimal Client will not be as good as if it was trained on its entire local dataset" The first half of this sentence directly contradicts the second half of it.
- The results in L305-339 are not related to the actual performance of the model for the clients. The so-called performance in L315 does not have an interpretable meaning in the FL context.

4. Experiments have no comparison to other RL baselines. While it is helpful to see that the algorithm can work with different aggregation or local training schemes, there is no comparison to other RL-based selection/adjustment baselines (e.g., Wang et al., 2020, Nguyen et al., 2022).

Ref:

- Wang, H., Kaplan, Z., Niu, D. and Li, B., 2020, July. Optimizing federated learning on non-iid data with reinforcement learning. In *IEEE INFOCOM 2020-IEEE conference on computer communications* (pp. 1698-1707). IEEE.
- Nguyen, N.H., Nguyen, P.L., Nguyen, T.D., Nguyen, T.T., Nguyen, D.L., Nguyen, T.H., Pham, H.H. and Truong, T.N., 2022, August. Feddrl: Deep reinforcement learning-based adaptive aggregation for non-iid data in federated learning. In *Proceedings of the 51st International Conference on Parallel Processing* (pp. 1-11).

Minor comments

- L116: the model parameter is a scalar?
- Use citet and citep properly.
- Capitalization. Why sometimes Federated Learning and sometimes federated learning? When shall we capitalize, or not? It would make more sense to have a consistent rule throughout the paper.
- L190 uses $\hat{f}$ while L191 is simply $f$
- Eq.(3) what does the $\Rightarrow$ indicate? How about Eq.(4)? Please use conventional notations for better readability.
- T in Eq.(8) should be t?

**Questions:**

Please address the questions in point number 2 in the weakness section.

---

### Official Review · Reviewer_Nbva · 2024-11-03

**Soundness:** 2
**Presentation:** 2
**Contribution:** 1
**Rating:** 3
**Confidence:** 3

**Summary:**

This paper proposes a novel framework that leverages deep reinforcement learning (DRL) to optimize client training data selection in federated learning (FL). The key idea is using a DRL agent to determine the optimal amount of training data for each client without oversharing information. Experiments on CIFAR-10/100 and FashionMNIST demonstrate potential benefits in personalization and overall performance. However, after careful review, I recommend rejection due to several critical issues.

**Strengths:**

+The authors' core insight about efficient data usage in FL clients is valuable, as reducing unnecessary data sharing while maintaining performance is an important practical concern in real-world FL deployments.

+Despite limitations, the paper provides extensive experiments across multiple standard datasets (CIFAR-10/100, FashionMNIST) and various FL aggregation methods (FedAvg, FedMedian, FedProx, etc.), demonstrating the framework's applicability across different settings.

**Weaknesses:**

1. Theoretical Foundation and Claims:

a. The paper's core claim that "training a client locally on more data than necessary does not benefit overall performance" lacks rigorous theoretical justification. The provided proof in Section 4.4 about performance bounds only shows a trivial bound without connecting to the optimality of data selection. In FL literature, there are many more advanced aggregation techniques addressing the imbalanced data problem (e.g., weighted aggregation). How does the proposed method work for different aggregation schemes?

b. Privacy preservation claims are made without formal analysis: i) no analysis or discussion of potential information leakage through the DRL agent's actions; ii) when the agent observes client performance, what are the implications to privacy? discussion is missing.

c. The paper proposes performance bounds but the bound $\sigma = \pi \omega_c$ is a trivial result as it just states performance difference is bounded by area difference. Neither further analysis of convergence properties for the DRL training nor theoretical guarantees about the optimality of learned policies is provided, which is critical to assess the significance.

2. methodological issues:

a. The reward function design in eq. (8) seems arbitrary. what are the intuitions of this reward design choice?

b. similarly, the two proposed action selection strategies should be justified

3. The experimental setup is limited:

a. all figures show no statistical significance

b. Only ResNet50 is used for experiments. is the proposed method generalized to different network architecture?

c. Comparison with other data selection strategies (such as [1] or simply weighted average) is missing

4. Missing Related Work and Context:

a. The paper overlooks crucial related work in FL data selection and client optimization. Notably: 1) methods for client data pruning/selection in FL; approaches using reinforcement learning for FL client selection, e.g, [1]; recent work on personalized FL with data filtering

b. without comparison to the related work, several claims about novelty need better contextualization: i) other works have explored adaptive data selection in FL; ii) the use of DRL for FL optimization is not new

---

[1] Rjoub, G., Wahab, O. A., Bentahar, J., Cohen, R., & Bataineh, A. S. (2024). Trust-augmented deep reinforcement learning for federated learning client selection. *Information Systems Frontiers*, *26*(4), 1261-1278.

**Questions:**

pls see Weaknesses section.

---

### Note · Authors · 2024-11-15

I have read and agree with the venue's withdrawal policy on behalf of myself and my co-authors.